# Asparagus (*Asparagus officinalis* L.) Root Distribution: Cultivar Differences in Mature Plantings

**Daniel Drost**

Department of Plants, Soils and Climate, Utah State University, 4820 Old Main Hill, Logan, UT 84322, USA; dan.drost@usu.edu

**Abstract:** Annual plant growth patterns and seasonal conditions have both been shown to influence asparagus (*Asparagus officinalis* L.) root development over time. Root biomass and distribution changes in mature asparagus cultivars are herein illustrated and described. Asparagus root length density and biomass were estimated from soil cores using a systematic field sampling approach each spring. Soil cores (0.9 m deep) were divided into 0.15 m lengths and fleshy roots collected from the soil. Root length density and dry weights were determined and root distribution maps generated from collected data. As asparagus plantings matured, the sampling year had a significant influence on root development. Fleshy roots grew deeper into the soil each year but the majority of roots of Atlas, Guelph Millennium, and Jersey Giant were found in the upper 60 cm of the soil profile. For the three cultivars evaluated, minor differences in root length and root weight occurred. By Year 6, Atlas showed a decrease in root length and weight when compared to Guelph Millennium and Jersey Giant. While spear yield differences between the varieties were not significant, Atlas tended to produce more very large and large spears compared to Guelph Millennium and Jersey Giant. These results increase our understanding of asparagus root development.

**Keywords:** root sampling; rooting depth; Atlas; Guelph Millennium; Jersey Giant





## 1. Introduction

An asparagus plant consists of the below-ground crown and above-ground fern. The crown includes the rhizome (underground stem), a mass of fleshy (adventitious) storage roots, and numerous apical buds [1,2]. The storage roots are initiated below or to the side of the rhizome and extend downward and/or outward [3]. The apical buds produce edible spears, and spears grow into fern later in the season. Healthy fern produce the CHO's needed to replace the energy used to grow spears and fern and extend the rhizome and grow fleshy roots [3,4]. Therefore, a large crown is key to plant longevity and productivity [3]. Wilson et al. [5] noted that the root system drives crop performance and that root CHO levels help determine productivity [5–8].

In young, newly established asparagus plantings, new fleshy root growth starts soon after initial spear emergence [2,9,10]. The F1 hybrid, UC157, grew bigger crowns (more fresh weight and more fleshy roots) than the open-pollinated cultivar UC800 [10,11]. Fleshy-root production in both cultivars began 6 to 10 weeks after emergence; however, there was little root production after 18 weeks in UC800, while UC157 continued to produce fleshy roots until late October. Differences in root initiation and production were noted between direct seeded and transplanted asparagus when evaluated [2]. Seeded plants produced 42 storage roots by the end of the growing season, while transplants produced 143 roots. Few new roots are initiated during spring spear and fern establishment [2,6].

In older, more mature plantings, new fleshy root production starts after fern establishment in the summer [12–14]. While yearly changes in fleshy and fine root production and their associated distribution patterns have been reported [12,15,16], few long term studies have been reported. There are reports of asparagus storage root CHO level variations

which are dependent on the size of the root system [5,14]. Consequently, the ability of asparagus to produce many large roots and to accumulate adequate CHO is crucial for high spear yields in both the current and subsequent harvests.

Recent [17] assessments of root growth in three asparagus cultivars over the first three growing seasons have been reported. It was noted that the cultivars Atlas and Jersey Giant extended fleshy roots further from and deeper into the soil when compared to the cultivar Guelph Millennium. However, there were few differences in total root mass or root length. This is one of the first reports on root changes over many years for cultivars with different growth habits. Others have speculated that drought tolerance [16,18], spear productivity potential [19–21], and soil-management approaches [13,22,23] are related to the cultivar (and potentially the root mass).

Long-term studies on asparagus root growth are needed. These studies will help to establish how quickly root systems develop. This information may then be used to improve crop nutrient and water management, practices that impact asparagus growth and spear productivity. Preliminary studies during the first three crop establishment years [17] for three asparagus cultivars show different rooting patterns. In this paper, we provide continued evaluation of the changes in fleshy root growth and distribution for these three asparagus cultivars as they become mature and enter full production potential.

## 2. Materials and Methods

This study was conducted at the Utah State University Greenville Research Farm, Logan, Utah, from 2016 to 2021. Seasonal climate conditions, site soil characteristics, soil test results, plant propagation methods, site preparation, plant spacing and arrangements, and yearly site management were similar each year and were fully reported in an earlier paper [17].

Three asparagus cultivars, Atlas F1, Guelph Millennium, and Jersey Giant, were evaluated. Atlas is a large spear cultivar with some heat tolerance. Guelph Millennium is a late season, medium-sized spear producing cultivar, and is cold-adapted. Jersey Giant is an early season, large-spear variety that tolerates cold winters.

Asparagus root sampling occurred in late April or early May of 2019, 2020, and 2021. The three cultivars were assessed by soil coring [13,17,24] and collected root data were used to determine the fleshy root length, biomass, and rooting density. Fleshy storage roots were field-collected [17], stored for up to 3 weeks [24], rinsed, surface-dried, and measured (total fleshy root fresh weight and root length, and average root diameter) before oven drying. Fleshy root length density (FL-RLD) was calculated as:

$$\text{FL-RLD} = L/V \ (\text{m} \cdot \text{m}^3), \tag{1}$$

where L is the sum of the fleshy root length (m) and V is the volume (m$^3$) of the soil core.

The spear harvest duration from 2019 to 2021 was the full season but varied depending on the yearly seasonal conditions. In 2019, there were 29 cuts (19 April–31 May; 7 frost events); in 2020, there were 31 cuts (25 April–30 May; 3 frost events); and in 2021, there were 36 cuts (17 April–1 June; 3 frost events). Harvest (cuts) duration is dependent on seasonal weather conditions and no harvests occurred on Sunday. Spear harvest locally concludes in early June to allow fern establishment and the root carbohydrate recharge required for short-growing-season production areas [25]. Harvested spears were graded to US standards for green asparagus. The spear size categories were very large (+22 mm diameter), large (18–22 mm), medium (13–18 mm), small (8–13 mm), very small (4–8 mm), or culls (<4 mm, bent/open heads, damaged, or unmarketable). Spears were weighed, trimmed, sized, counted, and then re-weighed. Total marketable productivity was the sum of all marketable size grades. In mid-summer (August), fern growth characteristics were evaluated. The average fern height and total stem per meter in three random locations in each plot were assessed.

Root parameters were analyzed by an analysis of variance to determine the main effects and interactions of year, cultivar, sampling depths, and location. The general linear

model procedure (SAS Institute, Cary, NC, USA) was used for the analysis of variance. Root distribution graphs were generated using a contouring program in Surfer 13 (Golden Software, Inc. http://www.goldensoftware.com, (accessed on 4 November 2022)). The effect of year was not analyzed for asparagus performance (yield and fern parameters) due to seasonal differences in start/stop dates and harvest duration.

## 3. Results

### 3.1. Fleshy Root Growth

The estimated root length and root weight for 2019–2021 are reported in Table 1. While there was no difference in total root length between Atlas, Guelph Millennium, and Jersey Giant in any given year, the sampling years ($p$ = 0.000) were significantly different from each other. In 2019, the total fleshy root length averaged 682 m/m$^3$ across the varieties, which was an 89% increase from 2018 (see [17]). The root length increased to 1092 m/m$^3$ in 2020 (60% increase), and it was 1114 m/m$^3$ in 2021 (2% increase). The total root fresh weight estimates between the three varieties evaluated were not different in 2019 and 2020. However, in 2021, the estimated root weights for Atlas and Jersey Giant were significantly lower than for Guelph Millennium. The root fresh weight for Atlas from 2018 to 2020 increased by 52% and 102%, respectively, but then decreased by 41% in 2021. In contrast, the root weight for Guelph Millennium increased by 29%, 60% and 60% from 2018 to 2021, respectively, while Jersey Giant increased by 85%, 13% and 21%, respectively, for the same period. The sampling years for the root fresh weight were significantly different ($p$ = 0.001) from each other. In 2019, the average fleshy root fresh weight was 7.20 kg/m$^3$ across the three varieties (54% increase from 2018 (see [17])). This increased to 10.91 kg/m$^3$ in 2020 (52% increase), and it was 13.37 kg/m$^3$ in 2021 (23% increase).

**Table 1.** Yearly changes in estimated fleshy root length (m/m$^3$), fresh weight (kg/m$^3$) and percentage change in root weight for the asparagus cultivars Atlas, Guelph Millennium, and Jersey Giant. Root growth was evaluated in April or May of 2019–2021.

| | 2019 | 2020 | 2021 |
|---|---|---|---|
| | Fleshy Root Length (m/m$^3$) | | |
| Atlas | 572.0 | 1130.4 | 833.0 |
| Guelph Millennium | 639.2 | 1063.2 | 1341.1 |
| Jersey Giant | 781.4 | 1083.2 | 1169.0 |
| LSD 0.05 | ns | ns | ns |
| | Fleshy Root Fresh Weight (kg/m$^3$) | | |
| Atlas | 5.592 | 11.273 | 6.688 |
| Guelph Millennium | 7.153 | 11.418 | 18.250 |
| Jersey Giant | 8.858 | 10.031 | 12.175 |
| LSD 0.05 | ns | ns | 5.817 |
| | % Change in Root Weight from Prior Year | | |
| Atlas | 52 * | 102 | −41 |
| Guelph Millennium | 29 * | 60 | 60 |
| Jersey Giant | 85 * | 13 | 21 |
| average | 55 | 58 | 14 |

ns = not significant; the main effect of year was highly significant for root length ($p$ = 0.008) and root fresh weight (0.021), while the year by variety interaction was not significant for root length ($p$ = 0.998) or root fresh weight (0.981). * = % change in root weight; reference [17].

Significant differences were not evident for total root length between the three asparagus varieties, but unique root distribution patterns were noted (Figure 1; Table 2). As Atlas matured, the FL-RLD was uniformly distributed over depth and distance from the

planted row. Widely spaced isolines indicate gradual changes in root length, while closely spaced lines indicate rapid changes in root growth. As Atlas aged from 2019 to 2021, the increase in RLD expanded outward and downward with a large concentration of roots near the asparagus crown. While there was a significant decrease in the root fresh weight for the cultivar Atlas from 2020 to 2021, RLD differences were between these two years were not evident. In Guelph Millennium, the FL-RLD was highly concentrated (closely spaced isolines) near the crown, and most roots were located near the soil surface. These patterns appeared the same from year to year, and the increasing number of isolines indicated significant increases in RLD and root mass as noted in Table 2. For Jersey Giant, the FLD was concentrated near the crown in 2019 and the isolines were closely spaced, indicating large changes with depth and distance. In 2020 and 2021, the RLD indicates continued outward and downward expansion with isolines situated farther apart. These patterns illustrate continued root growth but at a slower rate as the plants became more mature as noted in Table 1.

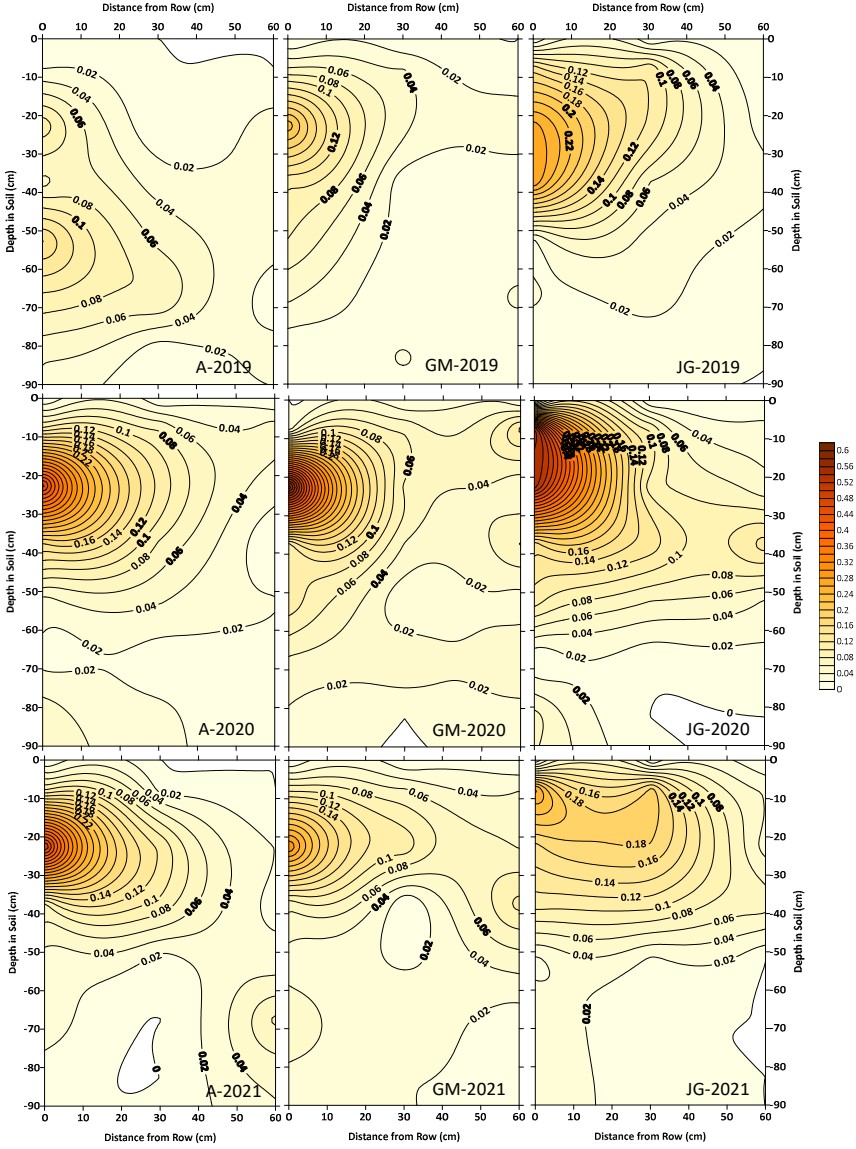

**Figure 1.** Changes in fleshy root length density (FL-RLD) from 2019 to 2021 for the asparagus cultivars Atlas (A), Guelph Millennium (GM), Jersey Giant (JG). Isolines represent the change in RLD (cm/cm$^3$) over depth (0–90 cm) and distance (0–60 cm) from the row.

**Table 2.** Percentage of root mass distribution by depth and distance and associated analysis of variance for asparagus cultivars (Atlas, Guelph Millennium, Jersey Giant) during 2019–2021. Plants were planted in April 2015, and root growth was assessed annually in late April or early May when the plants were mature and in full production.

| | **Distance from the Row (cm)** | | | | | | | | |
| | **0** | **30** | **60** | **0** | **30** | **60** | **0** | **30** | **60** |
| | 2019 | | | 2020 | | | 2021 | | |
| **Depth (cm)** | Atlas | | | | | | | | |
| 0–30 | 33% | 18% | 4% | 59% | 12% | 5% | 49% | 17% | 3% |
| 31–60 | 10% | 17% | 5% | 10% | 7% | 2% | 12% | 6% | 3% |
| 61–90 | 9% | 3% | 2% | 2% | 2% | 1% | 5% | 3% | 1% |
| | Guelph Millennium | | | | | | | | |
| 0–30 | 63% | 7% | 3% | 63% | 12% | 7% | 68% | 12% | 4% |
| 31–60 | 12% | 5% | 3% | 8% | 5% | 2% | 9% | 1% | 3% |
| 61–90 | 4% | 2% | 1% | 0% | 3% | 0% | 1% | 1% | 1% |
| | Jersey Giant | | | | | | | | |
| 0–30 | 59% | 13% | 1% | 58% | 11% | 5% | 59% | 17% | 5% |
| 31–60 | 8% | 8% | 2% | 13% | 6% | 2% | 9% | 4% | 2% |
| 61–90 | 3% | 3% | 2% | 4% | 0% | 0% | 2% | 1% | 0% |
| Source | df | MS | *p*-value | MS | *p*-value | | MS | *p*-value | |
| Reps | 3 | 0.00002 | 0.576 | 0.00012 | 0.502 | | 0.00013 | 0.778 | |
| Variety (V) | 2 | 0.00013 | 0.085 | 0.00008 | 0.764 | | 0.00024 | 0.601 | |
| Error a | 6 | 0.00003 | | 0.00014 | | | 0.00036 | | |
| Location (L) | 2 | 0.42661 | 0.000 | 0.53381 | 0.000 | | 0.54032 | 0.000 | |
| V * L | 4 | 0.02595 | 0.006 | 0.00063 | 0.963 | | 0.00655 | 0.451 | |
| Depth (D) | 2 | 0.41977 | 0.000 | 0.71596 | 0.000 | | 0.71806 | 0.000 | |
| V * D | 4 | 0.01514 | 0.069 | 0.00186 | 0.779 | | 0.00747 | 0.382 | |
| L * D | 4 | 0.26859 | 0.000 | 0.37152 | 0.000 | | 0.35011 | 0.000 | |
| V * L * D | 8 | 0.02051 | 0.021 | 0.00069 | 0.958 | | 0.00817 | 0.333 | |
| Error b | 83 | 0.00669 | | 0.00418 | | | 0.00704 | | |

ns = not significant; the main effect of year was highly significant for root length (*p* = 0.008) and root fresh weight (0.021), while the year by variety interaction was not significant for root length (*p* = 0.998) or root fresh weight (0.981).

While the root distribution patterns look similar for the three asparagus varieties evaluated, there are important differences (Table 2). In 2019, there was a significant variety by location by depth interaction. Atlas (33%) had fewer roots in the 0–30 cm depth and in the planted row (0 cm distance) than either Guelph Millennium (63%) or Jersey Giant (59%). Atlas also had more roots in the 30 cm distance and 0–30 cm and 31–60 cm depths than Guelph Millennium or Jersey Giant. As depth and distance increased, fewer differences in root percentage were noted. In 2020 and 2021, the variety by depth or distance interactions were no longer significant. In all years, there was a significant location by depth interaction. As the distance from the row (0, 30, 60 cm locations) and depth (0–30, 31–60, and 61–90 cm locations) increased, the percent fleshy root mass decreased for each of the three cultivars evaluated.

### 3.2. Crop Productivity

In the yearly assessment of fern number (*p* = 0.530) and height (*p* = 0.944), no differences in growth were noted between the three asparagus cultivars (data not shown) and there

was no difference between years. Each year, plants produced 40–45 stems per m of each row with an average fern height of 1.55–1.60 m.

The total marketable yield was not different between Atlas, Guelph Millennium, or Jersey Giant in 2019, 2020, or 2021 (Table 3). The total number of cuts each year and the starting dates were different; therefore, the years were analyzed separately. In 2019, Atlas produced significantly more very large spears compared to Guelph Millennium and Jersey Giant. However, the yield of small and very small spears was significantly higher for Guelph Millennium compared to Atlas or Jersey Giant. In 2020, there were no differences in yield for any spear size class between the three cultivars evaluated. In 2021, Atlas again produced significantly more very large and larger spears compared to Guelph Millennium, but was not different from Jersey Giant. In 2021, Guelph Millennium did produce more small spears compared to Atlas and Jersey Giant.

**Table 3.** Spear productivity differences for Atlas, Guelph Millennium, and Jersey Giant (kg/ha) from 2019 to 2021. Spear harvest length varied in 2019, 2020, and 2021 due to differences in environmental conditions during the spring.

| Variety | Total Marketable | Very Large (+22 mm) | Large (18–22 mm) | Medium (13–18 mm) | Small (8–13 mm) | Very Small (4–8 mm) |
|---|---|---|---|---|---|---|
| **2019 Spear Yield (kg/ha)—(6-Week Harvest (19 April–31 May); 29 Cuts)** | | | | | | |
| Atlas | 1964 | 217 | 457 | 689 | 493 | 108 |
| Guelph Millennium | 2687 | 20 | 142 | 831 | 1214 | 480 |
| Jersey Giant | 1866 | 41 | 270 | 721 | 689 | 146 |
| LSD 0.05 | ns | 156 | ns | ns | 597 | 179 |
| **2020 Spear Yield (kg/ha)—(5-week harvest (25 April—30 May); 31 cuts)** | | | | | | |
| Atlas | 3001 | 146 | 248 | 1022 | 1114 | 471 |
| Guelph Millennium | 3604 | 84 | 608 | 1432 | 1084 | 397 |
| Jersey Giant | 3645 | 172 | 427 | 1552 | 1072 | 421 |
| LSD 0.05 | ns | ns | ns | ns | ns | ns |
| **2021 Spear Yield (kg/ha)—(6-week harvest (17 April–1 June); 36 cuts)** | | | | | | |
| Atlas | 3650 | 141 | 1071 | 1657 | 712 | 69 |
| Guelph Millennium | 4135 | 0 | 427 | 1849 | 1752 | 107 |
| Jersey Giant | 3494 | 23 | 740 | 1808 | 856 | 67 |
| LSD 0.05 | ns | 131 | 494 | ns | 268 | ns |

ns = not significant; the main effect of year was not analyzed due to differences in harvest duration and starting dates that were influenced by year-to-year differences. Total marketable yield is the sum of the difference size classes.

## 4. Discussion

The study of asparagus roots is extensive and centers on aspects related to rooting depth, root distribution through the soil profile, and root age [13,15,26–28]. Most studies have evaluated fleshy storage roots. These are easy to identify, grow quite long (1–2 m), are long lived (6 years) [26], and are important for carbohydrate storage [12,14,15,28]. Less information is available on the growth of fleshy roots over time [14,17] or how different asparagus varieties compare in root development [11,29]. Our preliminary findings showed that root growth for the three varieties changes over several years [17]. When asparagus plants are young, roots grow downward and outward from the crown as the root system expands. Initially, total root length and root mass were not different between the three cultivars evaluated [17]. We noted differences in root distribution patterns that may be important when making crop-management decisions during the establishment years. Both

Atlas and Jersey Giant grew fleshy roots that extended further out from the crown and deeper into the soil compared to Guelph Millennium. Deeper rooting may be important particularly in lighter sandy soils with a low water-holding capacity. Differences in water-use between the cultivars Jersey Supreme and Guelph Millennium have been reported [18], with Jersey Supreme tolerating drought better than Guelph Millennium. Cultivars with a more expansive, larger root system can access water more efficiently and may thus tolerate drought conditions better. Differences in root mass for Gijnlim and Guelph Millennium have been noted in the first years after planting [29]. The Guelph Millennium rooting pattern was denser and shallower in the soil profile [29], similar to those reported in our earlier work [17].

As the three asparagus varieties continued to mature, the fleshy root length and root weight (Table 1) continued to increase. The root length and mass were not different between Atlas, Guelph Millennium, and Jersey Giant in 2019 or 2020, but by 2021, six years after planting, the fleshy root weight of Atlas showed a significant decrease compared to Guelph Millennium or Jersey Giant. The root weight (and length) loss in Atlas is not fully understood. Cultivar adaptation to local conditions may partially explain the decline [25,30–32]. Atlas is a cultivar bred for and adapted to hotter desert-like growing conditions and thus may be less adapted to the cold production region used in this study. In contrast, both Guelph Millennium and Jersey Giant are reported to be cold tolerant and their continued root size development (Table 1; Figure 1) each year suggests they are well adapted to the environmental conditions of the study area. Additional work is required on cultivar adaptation and root development over longer time periods to determine whether root length and weight changes continue to occur. To our knowledge, only one other study has looked at root changes over longer time periods [14]. They reported the root biomass accumulation over the initial three years after planting. Given the longevity of asparagus, having detailed root distribution information along with the dynamic changes that occur from year to year may help to estimate the nutrient storage capacity [4,23], improve irrigation efficiency [16,18], and estimate the root carbohydrate reserves [14,29], all of which are important for asparagus growth and productivity.

Total spear productivity was similar between the three cultivars evaluated each year (Table 3) and there were few differences in productivity between the three years. The production year 2019 was quite cold and there were seven frost events which damaged emerging spears and interrupted the spear harvest. In 2020 and 2021, harvest weather conditions were better with fewer adverse temperatures. As in our earlier findings [17], Atlas produced more very large and large-diameter spears (Table 3). Atlas is well-known for producing large-diameter spears. In contrast, Guelph Millennium produced significantly more small and very small spears compared to Atlas or Jersey Giant (Table 3). Asparagus productivity is related to root carbohydrate storage [5–9,14], and root size can influence total CHO storage [13,29]. While root size differed among the three cultivars evaluated, their spear yield differences were not different. In mature asparagus fields [5,14], root masses of up to 1.2 kg dry weight/plant (approximately 1.0–10.5 kg fresh weight/$m^3$) have been reported. The root masses reported here (Table 1) are within that range. Root growth is commonly quite stable from Years 5–10 in the production cycle [30–32]; therefore, stable spear yields and fern biomass would be expected for the three cultivars in future years. While Atlas showed some reduction in root growth in Year 6, additional field sampling is required to determine whether this will have an impact on spear productivity or was a sampling anomaly. Year-to-year root growth differences may contribute to yield variability; thus, long term root mapping can help to identify these cultivar differences and could be used to assess crop management influences, how abiotic or biotic factors impact asparagus performance, or if specific cultural practices are beneficial to plant performance.

Studies conducted by Drost and Wilcox-Lee [12] and Drost and Wilson [13] evaluated root growth in one production season, for one asparagus cultivar, and in mature plantings. These studies clearly showed in-season changes in root growth, and identified distinct differences in root distribution and biomass, but did not compare different varieties or look

at root growth over many growing seasons. Our initial study [17], showed that young developing Guelph Millennium roots were more horizontal (shallow) and located in the upper regions of the soil profile in the first three years after planting. In contrast, Atlas and Jersey Giant root systems spread deeper and wider into the soil profile. Final root mass and root length differences were not noted during crop establishment for any of the cultivars evaluated [17]. In a continuation of that study, during Years 4–6 (Figure 1), fewer differences in rooting distribution were evident. For the three varieties evaluated, the bulk of the roots were located in the top 30 cm of the soil profile (Table 2) with some roots extending down to 90 cm. Soil surface tillage operations have been shown to reduce asparagus performance [33,34]. Tillage fills in the planting furrow and reduces weed pressure, but it also damages root growth near the soil surface [12,13]. Little or no root loss near the soil surface was noted over the duration of this project (Figure 1; Table 2) as the field was maintained as no-till. Others have reported on the impact of seasonal tillage in asparagus and recommend caution and careful monitoring of tillage operations to minimize root damage and productivity losses [14,21,33,34]. Simple root sampling, as shown in this study, is an easy way to evaluate whether management practices are influencing root development.

Root sampling over many years in this and the prior study [17] has demonstrated the growth and development patterns of asparagus. Paschold et al. [14] also noted similar rapid changes in asparagus root fresh weight (variety not identified) over the first three growing seasons after planting. Their findings showed a 230% increase in root weight between Year 1 and Year 2, followed by a 19% and 11% increase in Year 3 and Year 4, respectively. They did not, however, sample further into the growth cycle [14]. Average values for the three varieties used in our studies [17] and presented here show a similar increase during the early establishment years. As the planting matured into Year 6, the percent root mass increase became more variable (Table 1).

The increase in biomass and root length density over many years suggests that Guelph Millennium and Jersey Giant are still actively growing. In contrast, there was a 25% to 40% decrease in RLD and biomass in Atlas from 2020 to 2021 which requires additional evaluation. In most studies, root development patterns look the same throughout the season [12,13], while year-to-year differences (Figure 1 and Table 2) reflect active growth and root mass increases. Others have noted that seasonal (during the year) changes in root growth in mature asparagus are not apparent [12,13,33]. However, periodic sampling during establishment [17] and as plants transition to maturity accurately captures the year-to-year development in asparagus root growth. While yield differences between the years and cultivars were not evident, the loss in total root mass and flesh root length noted in Atlas in Year 6 (2021) warrants further evaluation. Future productivity losses may be expected if additional root loss occurs.

## 5. Conclusions

In conclusion, year-to-year sampling clearly illustrates the growth and distribution of fleshy asparagus roots and adequately reveals differences in the rooting patterns of the three asparagus cultivars studied. Secondly, root development patterns in younger establishing plants [17] are different from more mature plants. During Years 1–3 after planting, roots grow rapidly and the changes from year to year are large. As plantings mature, the percent change in growth slows. Identifying these changes can help asparagus growers and researchers to improve crop-management practices, thus ensuring field longevity and stabilized spear productivity. Finally, longer-term assessments of root growth and additional evaluations of other important cultivars of asparagus are required to determine when root losses begin to impact plant performance. If root development changes over time are similar to those in the cultivars evaluated in this study, then strategies can be developed to ensure plant longevity. This additional information will help to improve our understanding of the growth and yield physiology of asparagus.

**Funding:** This research was funded (UAES1685) by the Utah Agricultural Experiment Station, 4800 Old Main Hill, Utah State University, Logan, UT 84322-4800; journal paper number UAES #9696.

**Data Availability Statement:** Data are contained within the article.

**Acknowledgments:** The assistance with data collection and processing of James Frisby of Utah State University and our undergraduate students, Josh Martin, Andrew Bohannon, Emmalee Rolfe, and Evan Christensen was greatly appreciated.

**Conflicts of Interest:** The author declares no conflict of interest. The funders had no role in the design of the study; in the collection, analyses, or interpretation of data; in the writing of the manuscript; or in the decision to publish the results.

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
