# Peer review of "Asparagus (Asparagus officinalis L.) Root Distribution: Cultivar Differences in Mature Plantings"

_horticulturae, doi:10.3390/horticulturae9090979_

Round 1
Reviewer 1 Report
Overall, this manuscript does not provide particularly important additional information compared to previously published works. Essentially, the author reiterates in several parts that this information can be useful to farmers in improving agricultural practices, but there is no indication of how these practices could be improved based on the reported results.
Line 122: Where can I see what is claimed? I suggest reorganizing the tables so that the significant effects, as well as the main factors and their interactions, can be seen more clearly and immediately. There are many possible ways of presenting the data. Why not also include a mean comparison test?
Line 171: In Table 2, the analyzed variable should be at least mentioned in the table heading. This table is also difficult to interpret. It would be much simpler to include the effect of the main factors on the side and bottom, along with a significance test for mean comparison.
Lines 165-167: This statement is taken for granted. There is truly very little innovation.
There are 8 self-citations out of the 34 references reported
Author Response
Reviewer #1 suggests that there is no particularly new or useful information in this manuscript.
We beg to differ. There are essentially no studies on asparagus root development over such a long time period. Paschold [citation 14] references his work but does not provide much detail or specify its importance other than noting the need for knowing root biomass to help improve quantification of root carbohydrates. Our finding are useful from that perspective as continued root growth documentation helps us understand how large root system get and that there are strong varietal differences in biomass accumulation over many years. We have added details on how this information can be used. Granted our work is for one unique environment but since findings are in line with those report elsewhere [14] it may indicate that these changes are universal, though I hesitate to climb out on that limb lest we are wrong. Thus the need for continual work (which I have been part of)!
Comment referencing several of the tables. These are clear and well organized. We have included appropriate statistics throughout and they show that years are highly significant but there were few interactions (noted in text). Additional details are thus not all that helpful. A significant year effect just shows that they are different and would be expected as plants are actively growing.
RE Table 2. The details are required as there were significant Variety by Location by Depth interactions in 2019 and then only Location by Depth effects in 2020 and 2021. More details are reported in the text to address this. A comprehensive analysis of variance table helps see where the main effects and interactions are occurring. Changing this would be counterproductive.
I commonly hesitate to add too many self-citations. The literature on root growth in asparagus is somewhat limited and it has been my goal over many years to document the changes. There are no other researchers willing to take on this work and without additional scientists willing to make the effort, what is one to do. If not cited then it would appear that there is no work going on.
Reviewer 2 Report
The manuscript entitled “Asparagus (Asparagus officinalis L.) Root Distribution: Cultivar Differences in Mature Plantings” (ID Horticulturae-2498551) investigated the variation of root developments among three cultivars over continuous three years. Root length density and biomass were estimated and compared among cultivars and years. As the author stated, a similar study (Drost 2023ï¼› Reference 17 in the current paper) has been conducted during the early stage of the plant growth, from 2016-2018. I am just wondering what is the novelty of the current study, although the author clarify that it is scarce for root study in mature stage of this species. As the data are quite similar between two studies, the structure and organization are similar too. This is not a good presentation.
The author addressed the importance of CHOs in the Introduction, however, no such convey was done. In addition, soil resources were not related to the dynamics of roots during the study period, it seems necessary to fully understand the root behavior.
Moreover, in the Discussion, author tends to compare the current findings with those from earlier stage, however, no quantitive data were reported. A comprehensive comparison via a conceptual figure could be a good choice.
Finally, in Line 258-259, the author said “Simple root sampling, as shown in this study, is an easy way to evaluate if management practices are influencing root development.” However, in the previous study that the same method sed, the Conclusion said “Firstly, novel procedures are needed to collect roots since they are difficult to access.” I am confused.
Author Response
Reviewer Response #3
Our work over six years provides insights into the changes that may or may not occur. While visually the information may seem similar, when fully evaluated, there are things that begin to stand out. I have added statements about the rapid change in root mass accumulation which is similar to limited work by Paschold [14]. From an organizational perspective, one may expect some similarity in layout which aids in understanding how the root system develops. It provides a better picture of changes over time which my and others earlier work failed to address (these early works looked at only a snapshot based on one year findings.
We did not address CHO as this was outside the scope of the grants used to support the project. Others [5-9,14] have adequate addressed this and some have suggested the need to know root mass (root size) to improve our understanding of total CHO load.
We added some additional details into Table 1 to quantify the changes in biomass accumulation and expounded on this in the text. I have run into issues with editors about re-use of data and thus I was hesitant to dig back into the earlier paper [17] and repeat the use of numbers. Your point is well taken and is appreciated.
I do not see these two statements at odd with each other. The coring system is easy (novel), does a good job of estimating the root parameters of interest, and if gathered, tell much about where roots are located, and provides insights into changes over time. Additional details were added to clear up confusion and how these could be use.